# Telerehabilitation Intervention in Transitional Care for People with COVID-19: Pre-Post Study with a Non-Equivalent Control Group

**DOI:** 10.3390/healthcare11182561

**Published:** 2023-09-16

**Authors:** Neuza Reis, Maria José Costa Dias, Luís Sousa, Filipa Canedo, Miguel Toscano Rico, Maria Adriana Henriques, Cristina Lavareda Baixinho

**Affiliations:** 1Nursing Research, Innovation and Development Centre of Lisbon (CIDNUR), 1900-160 Lisbon, Portugal; 2Centro Hospitalar Universitário Lisboa Central, 169-045 Lisboa, Portugal; cdias@chlc.min-saude.pt; 3Higher School of Atlantic Health, 2730-036 Barcarena, Portugal; luismmsousa@gmail.com; 4Portugal Comprehensive Health Research Centre (CHRC), 7000-811 Evora, Portugal; 5NOVA Medical School|Faculdade de Ciências Médicas (NMS|FCM), Universidade NOVA de Lisboa, 169-056 Lisbon, Portugal; filipa.canedo@gmail.com (F.C.); migueltoscanorico@gmail.com (M.T.R.); 6Lisbon Nursing School, Nursing Research, Innovation and Development Centre of Lisbon (CIDNUR), 1900-160 Lisbon, Portugal; ahenriques@esel.pt

**Keywords:** long COVID, telerehabilitation, dyspnea, fatigue, depression, transitional care

## Abstract

SARS-CoV-2 infection and its resulting sequelae have increased the prevalence of people with respiratory symptoms, with impacts on functional capacity, quality of life, anxiety, depression, and mental health. To mitigate this problem, one challenge has been the design and implementation of interventions that simultaneously allow for education, rehabilitation, and monitoring of people with long COVID, at a time when health services were on the verge of rupture due to the volume of people with active COVID and in need of intensive care. Telerehabilitation emerged as a mode for providing rehabilitative care that brought professionals closer to patients and enabled continuity of care. The present study aimed to evaluate the results of a telerehabilitation intervention for people with injuries associated with SARS-CoV-2 infection in hospital-community transitions, considering their degree of dependence in performing activities of daily living, respiratory symptoms, fatigue, gait capacity, muscle strength, and experience with anxiety and depression. A pre-post study with a non-equivalent control group was carried out with a total of 49 participants (intervention group n = 24; control group n = 25). The post-intervention results showed an increase in saturation, a decrease in heart rate, an improvement in the impact of post-COVID functionality, a decrease in fatigue, a decrease in perceived effort, and a decrease in depressive and anxiety symptoms. The telerehabilitation intervention, which combined educational strategies with respiratory and motor rehabilitation, helped improve global functionality and self-care, with clinical and functional impacts.

## 1. Introduction

In February 2020, the World Health Organization announced COVID-19 as the name of the novel coronavirus disease 2019, caused by the severe acute respiratory syndrome coronavirus 2 (SARS-CoV-2) [1]. COVID-19 is a highly infectious disease, which causes respiratory, physical, and psychological dysfunctions. About 80% of patients with COVID-19 present mild to moderate symptomology, and 20% develop severe or critical illness [2]. Among patients with symptomatic COVID-19, cough, myalgias, and headaches are the most reported symptoms. Other features, including shortness of breath, chest pain, fever, and fatigue, have also been well described. Pneumonia is the most frequent serious manifestation of infection, characterized primarily by fever, cough, dyspnea, and bilateral pulmonary infiltrates on chest imaging [3].

After COVID-19, patients may experience a long-term reduction in functional capacity, exercise tolerance, and muscle strength, regardless of their previous health status or disabilities, in addition to persistent radiographic manifestations and anxiety and depression [4,5,6]. Long COVID is a health condition that is still poorly understood, but it is recognized to have an attributable disease burden and is considered highly disabling with implications for social life and work capacity. A report presented by the World Health Organization (WHO) in the context of long COVID presented data showing a reduced workload among 45% of people and another 22% who did not return to work [7]. The National Institute for Health and Clinical Excellence (NICE) defines long COVID as a syndrome with signs and symptoms of COVID-19 that develop during or after infection, are present for more than 12 weeks, and cannot be better explained by other diseases [8].

The likelihood of developing long COVID is unrelated to the severity of the acute disease, although it is more common in hospitalized patients. The most frequent symptoms are fatigue, dyspnea, thoracalgia, generalized pain, myopathy, palpitations, cognitive dysfunction, memory changes, dysautonomia, gastrointestinal symptoms, altered sleep patterns, mood swings, symptoms of depression and anxiety, dry skin, and others. Patients may present with only one or multiple symptoms, which can be constant, transient, or fluctuating, and may change in nature over time [7,9,10,11,12]. Less often, it may present as a multisystemic disease with myocarditis and/or thromboembolic complications [9,13].

Pulmonary rehabilitation is a multidisciplinary intervention that includes exercise, education, and behavioral interventions aiming to reduce symptoms, optimize functional status, and improve quality of life through stabilizing or reversing systemic manifestations of diseases [14]. Potential chronic burden of symptoms and mental and physical dysfunction of COVID-19 raised the need for rehabilitation [15]. Evidence shows that respiratory rehabilitation improves symptoms of dyspnea, relieves anxiety, minimizes disability, preserves function, and improves quality of life both in the acute phase and after discharge [16].

After COVID-19 emerged and caused the collapse of health systems, many patients were not able to receive their treatments, because institution-based pulmonary rehabilitation programs were forced to significantly reduce enrollment, or in some cases completely shut down, during the pandemic [17]. Prior to the COVID-19 pandemic, telemedicine and telehealth did not gain significant traction or widespread adoption among healthcare professionals. However, the emergence of the pandemic highlighted the need for European authorities to encourage the utilization of these tools in everyday clinical practice, facilitating their effective implementation without obstacles [18]. Telerehabilitation offers an alternative for delivering rehabilitation services, and consists of providing rehabilitation services through telecommunication networks or the internet, offering remote treatments to patients from a distance [19]. The efficacy and safety of this digital remote rehabilitation have been proven non-inferior to traditional rehabilitation [20].

This study aimed to evaluate the results of a telerehabilitation intervention for people with injuries associated with SARS-CoV-2 infection in hospital-community transitions, considering their degree of dependence in performing activities of daily living, respiratory symptoms, fatigue, gait capacity, muscle strength, and experience with anxiety and depression.

## 2. Materials and Methods

### 2.1. Design

The methodological choice of a quantitative pilot pre-post with a non-equivalent control group study was justified because the major goals of this type of study are to assess the feasibility and to avoid the potentially disastrous consequences of embarking on a large study, which could potentially “drown” the whole research effort [21]. This type of study can range from evaluating the feasibility of protocol implementation to investigating the potential mechanisms of efficacy for a new intervention [22], allowing researchers to make decisions about larger-scale studies, and identify difficulties and needs over time. The study project is registered in the open science framework: https://doi.org/10.17605/OSF.IO/E8S6Y.

### 2.2. Setting and Participants

This study took place in a Portuguese university hospital center, in the region of Lisbon and Vale do Tejo. This hospital has been at the forefront in the screening and treatment of patients with COVID-19, both in the emergency department and for hospitalization in wards or intensive care. The inclusion criteria for the sample were: adults and older adults, of both sexes, hospitalized with SARS-CoV-2 infection, who presented a score on the Post-COVID-19 Functional Status Scale (PCFS) ≥ 3 (unable to perform activities of daily living); on the Functional Assessment of Chronic Illness Therapy Fatigue Scale (FACIT) ≤ 26 (moderate/intense); on the Borg dyspnea scale ≥ 4 (moderate/intense); and with saturations at rest of 94–95% and/or 92% on effort.

The sample was intentional, following the guidelines that indicate that, in general, sample size calculations are not required for some pilot studies [19]. The sample for a pilot must be representative of the target study population. It should also be based on the same inclusion/exclusion criteria as the main study. As a rule of thumb, a pilot study should be large enough to provide useful information about the aspects that are being assessed in terms of feasibility [11,12,13,14,15,16,17,18,19,20,21,22].

### 2.3. Intervention

The review of the literature and the recommendations of the World Health Organization regarding rehabilitation programs for people with long COVID informed the design of the rehabilitation program. The program consisted of ventilatory control training, aerobic exercises, muscle strengthening exercises, respiratory muscle training, and flexibility and balance training [1,4,7,8,9,13,23,24,25]. During the program, the progression of training intensity was adapted according to the perception of dyspnea according to the modified Borg scale [26].

The established respiratory rehabilitation program consisted of exercises at home supervised by video calls, which required telemonitoring. The program consisted of an intensive 12-week phase with interventions three times a week, and a maintenance phase of 2 weeks, with interventions once a week. The goal was to promote adherence to the therapeutic regimen, manage long-term oxygen therapy, teach breathing exercises, and promote the efficient use of resources [23,24].

The general objective of this strategy was to contribute to improved outcomes, enable patients to achieve a frequency of physical exercise of three to five times a week, correct deficits in health-related behaviors, and provide knowledge for them to develop strategies to deal with the constraints of everyday life.

### 2.4. Control Group

The participants allocated to the control group received the usual care: an initial clinical evaluation, management of the therapeutic regimen, education, and training relative to their health status. Although they had access to the telerehabilitation program, they were not part of the program because they refused or because they lacked digital literacy. The control group did not participate in the supervised rehabilitation program, receiving only physical activity education. After the end of the control period, they were given the chance to access the unsupervised exercise program.

### 2.5. Outcomes and Measures

In addition to data related to age, sex, comorbidities, and therapy, the following instruments were applied:PCFS [27]—assesses the functional capacity of the person post-COVID. This scale assesses post-COVID limitations in activities or tasks at home, school or work, as well as whether there was a need to change lifestyle after the illness [27].FACIT fatigue scale [28]—allows the self-perceived assessment of fatigue in the physical, functional, emotional, and social dimensions [26].One-minute Sit to Stand Test [29,30]—applied to evaluate endurance and muscle strength of the lower limbs, with a strong correlation with other stress tolerance measures.LCADL (the London Chest Activity of Daily Living scale) [31,32]—assesses dyspnea associated with the performance of life activities in four domains: personal care; household activities; physical activities; and leisure activities.mMRC (modified MRC Dyspnea questionnaire) [33]—rates the impact of dyspnea on daily activities.Borg scale [34,35]—evaluates subjective perceived effort, dyspnea, and/or fatigue of individuals subjected to a certain level of physical effort.HADS (Hospital Anxiety and Depression Scale) [36,37]—this has two domains, one to assess anxiety and the other for depression.Six-minute gait test [38]—assesses gait quality and functional physical capacity.

### 2.6. Data Collection

The study took place between November 2021 and January 2022. Participants were evaluated in person before the start of the rehabilitation program and again at the end of the program. These assessments took place in a post-COVID-19 outpatient consultation in a Portuguese university hospital center.

In both in-person meetings, all signs and symptoms associated with the patients’ health condition in the long COVID context, such as hemodynamic stability, dyspnea assessment, dyspnea assessment associated with activities of daily living and personal care, fatigue, anxiety and depression, functionality assessment, and impact of signs and symptoms on quality of life were evaluated. The evaluation was performed at both times by the rehabilitation nurse who carried out the program.

The rehabilitation program was based on the recommendations of the European Respiratory Society (ERS) [39], and patient education was based on WHO guidelines for Support for Rehabilitation: Self-Management after COVID-Related Illness [40]. The program lasted 12 weeks, with three sessions per week, followed by a maintenance phase of two weeks, with one session a week, for a total of 38 supervised sessions each lasting approximately 60 min. Telerehabilitation sessions were held at the patients’ homes, near a steady chair and table, and supported via telemonitoring.

The program included a warm-up and flexibility training (10 min), balance training (5 min), aerobic training (5–20 min), muscle strength training (20 min), and a cool-down and flexibility training (5–10 min).

The first segment (warm-up and flexibility training) included mobility exercises, low-intensity aerobics, and flexibility exercises. Breathing control exercises, breathing with half-closed lips, diaphragmatic breathing, and dyspnea control positions were also incorporated. Flexibility training included stretching of the main joints of the body, such as the neck, shoulders, elbows, wrists, trunk, hips, knees and ankles. In each session, the muscle groups worked on in aerobic and muscle strength training were also stretched.

The balance exercises were tailored to the challenges faced by each patient in performing activities of daily living. They consisted of progressive exercises according to the general principles of balance training: acquiring postures with a progressively reduced support base, including dynamic movements that disturb the center of gravity; assessing which postural muscle groups are involved; reducing sensory input; and associating additional activities with the task of maintaining postural control.

As for aerobic training, at first, the exercise lasted 5 min, and throughout the program, it increased to 30 min. The modified Borg scale [34] was used to adjust the training load, aiming for the patient to present a score of between 4 (moderate) and 5 (somewhat difficult) during training.

During muscle strength training, elastic bands, ankle weights and free weights were used. Muscle strength training was moderately intense, with a perceived effort level of 5 on the modified Borg scale [34,35]. In each session, six or seven exercises were carried out with the main muscle groups. This included two sets of ten repetitions, with a 1-min interval between them. During the training, patients were instructed to maintain a regular breathing pattern, i.e., inhaling in the eccentric contraction phase and exhaling in the concentric phase. The professionals reinforced the teaching of exhalation through half-closed lips.

The training of the inspiratory muscles was carried out in an interval manner every day, in two blocks of 10 min, five times a week. The modified Borg scale was used to adjust the progression of inspiratory training load [34,35].

The program ended with low-intensity exercises, referred to in the warm-up phase.

Peripheral oximetry oxygen saturation and heart rate were monitored during the program.

### 2.7. Statistical Analysis

Statistical treatment was performed with SPSS^®^ (Statistical Package for the Social Sciences), version 25. Descriptive statistics (relative and absolute frequencies, mean and standard deviation) and inferential statistics with nonparametric tests were used for data processing. In all situations, the significance level adopted was *p* ≤ 0.05.

### 2.8. Ethical Considerations

This study was approved by the Ethics Committee of the Central University Hospital Center of Lisbon (CHULC) (Resolution no. 1209/2022 of 18 March 2022). All participants were asked to provide informed consent, after both oral and written explanation of the objectives of the study, clarification of the purposes of participation, times of participation, and the average time spent participating. The information was conveyed in accessible language and participants were allowed to ask questions, which were answered by the researcher responsible for data collection. They were guaranteed the right to withdraw from the study without any impact on their care provision and the rights and duties of patients in health care. The researcher who collected the data assigned a unique code to each participant, ensuring anonymity, because only he knew the correspondence between the code and the participant’s personal data. Confidentiality of the data was guaranteed.

## 3. Results

Forty-nine people were included in the study between November 2021 and the end of January 2022. The intervention group consisted of 24 participants, and the control group, 25. Participants in the control group were on average 63.9 years old (±9.2), and 64% were women. In the intervention group, the 24 participants were on average 50.2 years old (±13.7) and 58.3% were men.

Table 1 shows the differences between the control and intervention groups pre-intervention. At baseline, only three variables presented significant differences in the two groups. O_2_ saturation (SPO_2_) values were higher in the control group (*p* < 0.0001). The household tasks dimension score was higher in the intervention group (*p* = 0.020), i.e., the higher the score, the greater the limitation.

Post-intervention, the intervention group presented lower heart rate (*p* = 0.005); better post-COVID functionality (PCFS) (*p* < 0.0001); improvement in fatigue (*p* < 0.0001), dyspnea (*p* < 0.0001), ability to perform personal care (*p* = 0.034), and ability in leisure activities (*p* = 0.01); improvement in the stand and the Sit to Stand Test (STS) (*p* < 0.0001); and lower perception of physical exertion (*p* < 0.0001) during the STS test (*p* < 0.0001), but increased HR during the STS test (*p* = 0.001). The intervention group showed a decrease in anxiety (*p* = 0.004) and depression (*p* < 0.0001) compared to the control group (Table 2).

Intra-group (control) evaluation showed that there was a significant variation in SPO_2_ (*p* < 0.0001), improvement in post-COVID functionality (*p* = 0.021), decreased perceived effort (*p* = 0.045), and increased ability to perform physical activity (*p* = 0.042) and in the functionality evaluated by the Sit to Stand Test (*p* = 0.008) (Table 3).

In the intragroup evaluation (intervention group), all variables showed significant variations in values, except for the LCADL dimension, in the total impact on activities of daily living (Table 4).

At the intervention group post-intervention baseline, there was an increase in saturation (*p* < 0.0001), a decrease in HR (*p* = 0.039), an improvement in the impact of post-COVID functionality (*p* < 0.0001), a decrease in fatigue (*p* < 0.0001), a decrease in perceived effort (*p* < 0.0001), an improvement in the values on the LCADL scale in the dimensions personal care (*p* < 0.0001), household tasks (*p* < 0.004), leisure (*p* < 0.003) and physical activities (*p* < 0.001). There was a decrease in depression (*p* < 0.0001) and anxiety (*p* < 0.0001) symptoms. In terms of performance in the Sit to Stand Test, there was an improvement in execution (*p* < 0.0001), in addition to a decrease in perceived effort (*p* < 0.0001) during its execution. There was also an increase in heart rate (*p* < 0.018) and SPO_2_ (*p* < 0.0001) when performing the Sit to Stand Test.

The telerehabilitation intervention allowed for improved global functionality and self-care, and decreased perceived effort and fatigue, with an improvement in O_2_ saturations. It also led to a significant improvement in anxiety and depression symptoms.

## 4. Discussion

This study validated a supervised 12-week telerehabilitation program carried out with adults with long COVID symptoms at least six months after infection by SARS-COV-2.

The results are encouraging and may be a starting point for more wide-ranging studies in the context of supervised telerehabilitation interventions recommended by rehabilitation nurses in the management of long COVID symptoms. Supervised telerehabilitation with telemonitoring support is feasible and safe. There were no adverse events, and adherence was high, with a recruitment rate consistent with previous respiratory rehabilitation trials [41,42,43].

The telerehabilitation program resulted in improvements in global functionality, physical capacity, and self-care, reduced dyspnea and fatigue, improved O_2_ saturations. It also showed a significant improvement in anxiety and depression symptoms, corroborating the results of other studies, which have observed similar gains [41,42,43]. A study that applied a telerehabilitation program in primary health care concluded that it effectively improved physical capacity, quality of life, and symptoms in adult survivors of COVID-19 [42].

Functional physical capacity was evaluated using the 1-min STS, which is recommended for telerehabilitation programs, presenting itself as a good alternative when performing the 6-min gait test is not possible [42]. Its execution allows for the evaluation of the person while maintaining safety conditions with the use of a chair [30,43,44,45]. The results showed better performance in the functional test in all participants 12 weeks after the start of the program; however, the intervention group showed greater gains than the control group.

The interventions allowed significant gains in physical functionality. Compared to the control group, the post-intervention group showed better results (*p* < 0.001), with lower perceived effort after STS (*p* < 0.001). These results are in line with other studies carried out on people after COVID-19 [15,16,17,18,19,20,30,44,45,46,47].

A comparison of the pre- and post-intervention results shows significant gains (*p* = 0.008), with an increase of more than 2.5 repetitions, in the execution of the 1-min STS. The minimum difference considered clinically effective in persons with long COVID has not yet been established. However, in COPD, the value of 2.5 repetitions is considered a predictor of effectiveness [48]. Patient populations cannot be compared, but the results showed that the gains were six times higher than the results of another telerehabilitation program carried out in the community [42].

The data also show a reduction in fatigue and dyspnea. These are symptoms present in people with long COVID-19 [11,48], and their persistence leads to a significant decline in physical capacity and self-care performance [49]. The implemented program, based on education, training and exercise training strategies, managed to reduce dyspnea (mMRC) and fatigue (FACIT), produce gains in the ability to perform activities of daily living (LCADL) that support personal care, and had an impact on daily life as evaluated by PCFS, with decreased anxiety and depression. These results support other studies published during the pandemic [14,15,16,17,18,19,20,30,41,42,43,44,45,46]. Exercise training improves functional capacity, controls symptoms, and has a favorable impact on daily life and self-care.

The results of this study can be used to support the use of new technologies and demonstrate how these can be used as tools to provide cost-effective, safe, and more comfortable care for patients, avoiding unnecessary trips to the hospital. In Portugal, because of the reorganization of the health system and a sharp decrease in the number of people in the interior of the country, many patients with COVID had to be moved more than 100 km to receive care. Without this e-health modality, it would not have been possible to ensure continuity of care [30] and equity in their access to health services. However, it is important to be aware that there are some documented risks regarding the use of telemedicine, namely the unsustainability of the enormous amount of clinical data to analyze, compare and record in medical records and the impossibility of verifying the reliability of the patient’s self-assessment [18].

As recommendations for implementation in clinical practice, it is necessary to take into account when designing distance intervention protocols (telerehabilitation) the following elements: (a) clinical factors associated with the patient such as age, adherence, and degree of autonomy, (b) factors associated with the disease such as the acute or chronic stage, and also, (c) the presence of caregivers and the availability of appropriate information technology tools [18].

This study reinforces the importance of telerehabilitation programs that are tailored to the real needs of each patient. It is a flexible program that can be adjusted to the daily routine of each patient. The benefits of telerehabilitation intervention are evident, and they are identical to the results of face-to-face rehabilitation, as corroborated by the literature [9]. Future studies should explore the impact of the program on ensuring continuity of care [40,50].

## 5. Limitations

The limitations of the study are related to the method, although a non-equivalent control group pre-test/post-test design has the advantage of having two moments of evaluation, does not have neither a true control group, neither the random assignment of the participants which increase the risk of bias into the results of the study [51]. In addition, the intentional selection of the sample associate to the number of participants, limits the generalization of the results.

It was not possible to assess the degree of customer satisfaction, which could add value to the effective gains with the rehabilitation program.

## 6. Conclusions

This study evaluated the results of a telerehabilitation program that combines educational actions and an exercise program, based on the recommendations of ERS and WHO. The program lasted 14 weeks, totaling 38 supervised sessions (three sessions/12 weeks and one session/2 weeks). The results show that the interventions implemented and mediated by technology contributed to the improvement of symptoms (dyspnea and fatigue), increased functional capacity, decreased perceived effort, and increased LCADL scores in the personal care, domestic tasks, leisure, and physical activity dimensions. There was also a decrease in depressive and anxiety symptoms.

The telerehabilitation program contributes to recovery and reintegration into the community, with gains in autonomy and contributes to health-related quality of life.

The integration of telerehabilitation into healthcare systems enhances the accessibility, enabling a response for those individuals who do not have access to in-person rehabilitation programs and who, while maintaining conditions of safety and hemodynamic stability, seek support. Telerehabilitation offers a highly accessible service, with a significant emphasis on health literacy, and acts as a facilitator for altering detrimental behaviors. It enables education and training within the patient’s realm, linking exercises to everyday functionality, thereby augmenting long-term advantages for both the patient, their family, and healthcare providers.

The program has the potential to be safely replicated, promoting the improvement of clinical and functional indicators in people with long COVID.

The challenges posed by the pandemic have become a great opportunity to reorganize rehabilitation care for people with long-term COVID-19. The positive outcomes of this study serve as a driving force behind the implementation of personalized telerehabilitation programs, tailored to individuals dealing with complex diseases. These programs place a significant emphasis on comprehensive patient assessment and their intrinsic value.

## Figures and Tables

**Table 1 healthcare-11-02561-t001:** Differences between the control and intervention groups pre-intervention. Lisbon, 2022.

Variables	Group	N	M (SD)	Me (IQR)	Mann-Whitney Test
SPO_2_	Control	25	97.6 (1.3)	98 (1.5)	−4.992 (*p* < 0.0001)
	Intervention	24	93.9 (2.8)	94 (4)	
FC	Control	25	81.5 (12.5)	80 (20)	−0.140 (*p* = 0.889)
	Intervention	24	79.7 (12.8)	80 (14.7)	
PCFS	Control	25	2.8 (0.7)	3 (0.5)	−1.826 (*p* = 0.068)
	Intervention	24	2.5 (0.5)	3 (1)	
FACIT	Control	25	26.9 (8.5)	27 (12.5)	−1.023 (*p* = 0.307)
	Intervention	24	29.8 (7.3)	27.5 (8.7)	
mMRC	Control	25	2.2 (1)	2 (2)	−0.021 (*p* = 0.983)
	Intervention	24	2.2 (0.8)	2 (1)	
LCADL_CP (Personal Care)	Control	25	5.9 (2.8)	5 (1.5)	−1.369 (*p* = 0.171)
	Intervention	24	6.2 (1.9)	6 (3.7)	
LCADL_TD (Housework)	Control	25	7.7 (7.6)	4 (8)	−2.332 (*p* = 0.020)
	Intervention	24	11.5 (6.5)	10 (8)	
LCADL_L (Leisure)	Control	25	5.6 (2.6)	6 (4.5)	−1.897 (*p* = 0.058)
	Intervention	24	4.5 (1.6)	4.5 (2)	
LCADL_AF (Physical Activity)	Control	25	4.4 (1.8)	4 (2.5)	−0.919 (*p* = 0.358)
	Intervention	24	3.8 (0.9)	4 (1)	
LCADL_F (Final Impact)	Control	25	3.2 (0.7)	3 (0)	−0.128 (*p* = 0.898)
	Intervention	24	3.2 (1.3)	3 (0)	
HADS Anxiety	Control	25	9 (5.5)	9 (9)	−1.143 (*p* = 0.253)
	Intervention	24	7 (4.6)	8 (7.7)	
HADS Depression	Control	25	9.4 (5.1)	8 (7.5)	−1.073 (*p* = 0.283)
	Intervention	24	7.3 (3.7)	8 (5.7)	
STS	Control	25	17.5 (6.2)	17 (7)	−0.794 (*p* = 0.427)
	Intervention	24	18.9 (4)	17 (5.5)	
Borg_sts	Control	25	5.2 (1.6)	5 (3)	−1.351 (*p* = 0.177)
	Intervention	24	5.8 (1.7)	5.5 (2.8)	
FC_sts	Control	25	101.3 (16.4)	98 (28)	−0.671 (*p* = 0.502)
	Intervention	24	103.5 (22.8)	97 (37)	
SPO_2__sts	Control	25	95.9 (2.7)	96 (3	−2.091 (*p* = 0.037)
	Intervention	24	94 (3.5)	95 (6.5)	

**Table 2 healthcare-11-02561-t002:** Differences between the control and intervention groups pre-intervention. Lisbon, 2022.

Variable	Group	N	M (SD)	Me (IQR)	Mann-Whitney Test
SPO_2_	Control	25	96 (2.6)	96 (3)	−1.880 (*p* = 0.060)
	Intervention	24	97.3 (1)	97(2)	
FC	Control	25	83.6 (12.4)	82 (20)	−2.783 (*p* = 0.005)
	Intervention	24	73.9 (10.2)	73.5(10.3)	
PCFS	Control	25	2.5 (0.5)	3(1)	−5.874 (*p* = 0.005)
	Intervention	24	0.8(0.6)	1 (0.8)	
FACIT	Control	25	27.2(9.5)	30 (16.5)	−5.327 (*p* = 0.005)
	Intervention	24	43.3 (5.6)	44 (11)	
mMRC	Control	25	1.8 (1)	2 (0.5)	−4.133 (*p* = 0.005)
	Intervention	24	0.7 (0.5)	1 (1)	
LCADL_CP (Personal Care)	Control	25	5.3 (1.8)	5 (2)	−2.126 (*p* = 0.034)
	Intervention	24	4.5 (1.1)	4 (0)	
LCADL_TD (Housework)	Control	25	6.6 (6)	4 (7.5)	−1.399 (*p* = 0.162)
	Intervention	24	8 (5.4)	6.5 (5.3)	
LCADL_L (Leisure)	Control	25	5.1 (2.2)	5 (3.5)	−2.578 (*p* = 0.010)
	Intervention	24	3.5 (1.3)	3 (1)	
LCADL_AF (Physical Activity)	Control	25	3.6 (1.5)	3 (3)	−1.342 (*p* = 0.180)
	Intervention	24	3 (0.7)	3 (1)	
LCADL_F (Final Impact)	Control	25	3 (0)	3 (0)	0.000 (*p* = 1.000)
	Intervention	24	3 (0.8)	3 (0)	
HADS Anxiety	Control	25	8.6 (4.8)	9 (7.5)	−2.848 (*p* = 0.004)
	Intervention	24	4.8 (3.1)	5 (5)	
HADS Depression	Control	25	9.4 (4.9)	10 (8)	−3.568 (*p* = 0.005)
	Intervention	24	4.6 (3.2)	4 (5)	
STS	Control	25	21.1 (8.2)	20 (11)	−4.228 (*p* = 0.005)
	Intervention	24	33.1 (8.2)	33.5 (9.5)	
Borg_sts	Control	25	4.9 (1.4)	5 (2)	−4.700 (*p* = 0.005)
	Intervention	24	2.8 (1)	3 (1)	
FC_sts	Control	25	102 (17.2)	102 (28)	−3.387 (*p* = 0.001)
	Intervention	24	117.3 (9.5)	116 (13.3)	
SPO_2__sts	Control	25	95.8 (3.3)	97 (3)	−0.041 (*p* = 0.967)
	Intervention	24	96.6 (1.5)	97 (1)	

**Table 3 healthcare-11-02561-t003:** Variation between baseline variables at the end of the intervention in the control group. Lisbon, 2022.

Variable	N	M (SD)	Me (IQR)	Wilcoxon Signed Ranks Test
SPO_2_	25	−1.7 (2.0)	−2 (3.0)	−3.501 (*p* < 0.0001)
FC	25	2.1 (9.8)	0 (0)	−0.612 (*p* = 0.541)
PCFS	25	−0.32 (0.6)	0(1)	−2.309 (*p* = 0.021)
FACIT	25	0.3(7.8)	0 (9.5)	−0.103 (*p* = 0.918)
mMRC	25	−0.4 (0.9)	0 (1)	−2.008 (*p* = 0.045)
LCADL_CP	25	−0.6 (1.8)	0 (1.5)	−1.399 (*p* = 0.162)
LCADL_TD	25	−1,1 (5.7)	0 (4.5)	−0.966 (*p* = 0.334)
LCADL_L	25	−0.5 (2.2)	0 (2)	−1.089(*p* = 0.276)
LCADL_AF	25	−0.8 (1.9)	0 (2.5)	−2.028 (*p* = 0.043)
LCADL_F	25	−0.2 (0.7)	0 (0)	−1.732 (*p* = 0.083)
HADS Anxiety	25	−0.4 (2.5)	0(0.5)	−0.950 (*p* = 0.342)
HADS Depression	25	−0.04 (3.2)	0 (0.5)	0.000 (*p* = 1.000)
STS	25	3.6 (6.5)	4 (7)	−2.666 (*p* = 0.008)
Borg_sts	25	−0.2 (1.4)	0 (0)	−0.846 (*p* = 0.397)
FC_sts	25	0.7 (11.1)	0 (0)	−0.297 (*p* = 0.767)
SPO_2__sts	25	−0.1 (2)	0 (0)	−0.141 (*p* = 0.888)

**Table 4 healthcare-11-02561-t004:** Variation between baseline variables at the end of the intervention in the control group. Lisbon, 2022.

Variable	N	M (SD)	Me (IQR)	Wilcoxon Signed Ranks Test
SPO_2_	24	3.3 (2.7)	3 (5.0)	−4.041 (*p* < 0.0001)
FC	24	−5.9 (13)	−5.5 (16.7)	−2.069 (*p* = 0.039)
PCFS	24	−1.7 (0.6)	−2(1)	−4.382 (*p* < 0.0001)
FACIT	24	13.5(7.7)	11.5 (10.5)	−4.261 (*p* < 0.0001)
mMRC	24	−1.5 (0.6)	−1.5 (1)	−4.326 (*p* < 0.0001)
LCADL_CP	24	−1.7 (1.9)	−1.5 (2)	−3.518 (*p* < 0.0001)
LCADL_TD	24	−3.4 (5.4)	−2 (3.7)	−2.902 (*p* = 0.004)
LCADL_L	24	−1.0 (1.5)	−1 (2)	−2.977 (*p* = 0.003)
LCADL_AF	24	−0.8 (0.9)	−1 (0.3)	−3.346 (*p* = 0.001)
LCADL_F	24	−0.2 (1.2)	0 (0)	−0.632 (*p* = 0.527)
HADS Anxiety	24	−2.2 (2.7)	−2(4)	−3.171 (*p* = 0.002)
HADS Depression	24	−2.8 (2.8)	−3 (4)	−3.485 (*p* < 0.0001)
STS	24	14.3 (7)	12 (8.7)	−4.289 (*p* < 0.0001)
Borg_sts	24	−3 (1.9)	−3 (3.8)	−4.122 (*p* < 0.0001)
FC_sts	24	13.8 (27)	18 (31.8)	−2.373 (*p* = 0.018)
SPO_2__sts	24	3 (3)	2 (4)	−3.597 (*p* < 0.0001)

## Data Availability

Data are available only upon request to the correspondent author.

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
