# Peer review of "Telerehabilitation Intervention in Transitional Care for People with COVID-19: Pre-Post Study with a Non-Equivalent Control Group"

_healthcare, 2023, doi:10.3390/healthcare11182561_

Round 1
Reviewer 1 Report
The Article focuses on the use of telerehabilitation as an intervention for individuals with injuries associated with SARS-CoV-2 infection. Reis et al. highlight the challenges in providing education, rehabilitation, and monitoring to individuals with long COVID during a time when healthcare services were overwhelmed. This study highlights that telerehabilitation can be an effective approach to improving overall functionality and self-care for individuals recovering from SARS-CoV-2 infection.
I recommend accepting this article after MINOR REVISIONS.
1. Discuss any limitations or challenges encountered during the implementation of telerehabilitation. Addressing these limitations will provide insights into potential areas for improvement or considerations when implementing similar interventions in different contexts.
2. Consider discussing patient satisfaction or feedback regarding their experience with telerehabilitation. Including qualitative data or patient testimonials can provide a more comprehensive understanding of how this mode of care was perceived by those receiving it.
3. There is a lack of recent literature citations. In page 1, “COVID-19 is a highly infectious disease, which causes respiratory, physical, and psychological dysfunctions.( DOI: 10.3390/nu15153443)”; in page 2, lines 7-9, “Long COVID is a health condition that is still poorly understood, but it is recognized to have an attributable disease burden and is considered highly disabling with implications for social life and work capacity. (DOI: 10.3389/fpubh.2022.908757)”.
4. Discuss the implications of your findings for future research or clinical practice. Consider how telerehabilitation can be further optimized or integrated into existing healthcare systems to address the long-term effects of SARS-CoV-2 infection and improve patient outcomes.
Minor editing
Author Response
- Discuss any limitations or challenges encountered during the implementation of telerehabilitation. Addressing these limitations will provide insights into potential areas for improvement or considerations when implementing similar interventions in different contexts.
Thank you for your comment which led me to reflect once more on the applicability of the program.
In fact, no limitations were found in the execution of the program. The university hospital center where it was carried out has always been a good ally. From the point of view of customers and their families, they have always been open and willing to implement it.
- Consider discussing patient satisfaction or feedback regarding their experience with telerehabilitation. Including qualitative data or patient testimonials can provide a more comprehensive understanding of how this mode of care was perceived by those receiving it.
This is a quantitative study, which followed the protocol approved by the CHULC ethics committee, so the degree of satisfaction was not analysed.. I consider a limitation that will be added in the text
“ It was not possible to assess the degree of customer satisfaction, which could add value to the effective gains with the rehabilitation program.”
- There is a lack of recent literature citations. In page 1, “COVID-19 is a highly infectious disease, which causes respiratory, physical, and psychological dysfunctions.( DOI: 10.3390/nu15153443)”; in page 2, lines 7-9, “Long COVID is a health condition that is still poorly understood, but it is recognized to have an attributable disease burden and is considered highly disabling with implications for social life and work capacity. (DOI: 10.3389/fpubh.2022.908757)”.
Thank you for your careful review. has been revised and the proposed references have been added
- Discuss the implications of your findings for future research or clinical practice. Consider how telerehabilitation can be further optimized or integrated into existing healthcare systems to address the long-term effects of SARS-CoV-2 infection and improve patient outcomes.
The telerehabilitation program contributes to recovery and reintegration into the community, with gains in autonomy and contributes to health-related quality of life.
The incorporation of telerehabilitation in health systems increases the offer, allowing to respond to all those who do not have access to face-to-face rehabilitation programs and who present conditions of safety and hemodynamic stability feel accompanied. Telerehabilitation is a service of great proximity, with a strong focus on health literacy and facilitator in changing harmful behaviors. it allows teaching and training in the sphere of the patient, relating exercise to day-to-day functionality, enhancing long-term benefits for both the patient and his family and for health care providers.
The challenges posed by the Pandemic have become a great opportunity to reorganize rehabilitation care for people with long-term Covid-19. The beneficial results of the study are the lever in the implementation of personalized telerehabilitation programs, aimed at the person with a complex disease, with a strong focus on the person's assessment and value.

Reviewer 2 Report
The article is interesting and methodologically appropriate; however, the discussions appear scant, especially concerning the discussion and conclusions. The introduction could also be enriched with considerations regarding how the pandemic period has stimulated telemedicine. Meanwhile, the discussion could highlight some of the potential challenges and risks associated with the use of these nonetheless valuable remote approaches. In this regard, I would like to suggest referring to the content presented in the following article
Telemedicine as a medical examination tool during the Covid-19 emergency: The experience of the onco-haematology center of tor vergata hospital in Rome
Postorino, M., Treglia, M., Giammatteo, J., ...Cantonetti, M., Marsella, L.T. ,
Author Response
The article is interesting and methodologically appropriate; however, the discussions appear scant, especially concerning the discussion and conclusions. The introduction could also be enriched with considerations regarding how the pandemic period has stimulated telemedicine. Meanwhile, the discussion could highlight some of the potential challenges and risks associated with the use of these nonetheless valuable remote approaches. In this regard, I would like to suggest referring to the content presented in the following article
Telemedicine as a medical examination tool during the Covid-19 emergency: The experience of the onco-haematology center of tor vergata hospital in Rome
Postorino, M., Treglia, M., Giammatteo, J., ...Cantonetti, M., Marsella, L.T. International Journal of Environmental Research and Public Healththis link is disabled, 2020
Thank you very much for the improvement recommendations. Contributions were introduced from the reference that you indicated in the introduction and discussion section, when talking about the implications for clinical practice.

Reviewer 3 Report
This is a concise and well-written report of a pilot study to test telerehabilitation of 24 Long COVID-19 patients living in Portugal in comparison with 25 in a control group who did not receive the intervention.
The strengths of this paper are that in almost all cases the authors have focused on referenced published during COVID-19 to support their claims. As well, their tables and explanation of them are clear. The weaknesses are that there are a few aspects of the method that are not explained in sufficient detail with COVID-19 related references. As well, the limitations sections needs to be expanded on in relation to the resulting control group and the references redone according to MDPI style. The page by page suggested edits follow.
Page by page suggested edits.
Page 2
Change “A recent report presented by WHO” to “A recent report presented by the World Health Organization (WHO)”.
“After COVID-19 emerged and caused the collapse of health systems, many patients were not able to receive their treatments because institution-based pulmonary rehabilitation programs were forced to significantly reduce enrollment, or in some cases completely shut down, during the pandemic.”—please provide at least one peer reviewed reference to support the claims of this statement.
Page 3
Change “the World Health Organization” to “the WHO”.
If the control group included those who refused participation or lacked digital literacy can they truly be said to be a control group? They were, in effect, different from the group who participated in the pilot study in these ways mentioned. The authors need to make this point about the control group in their limitations section.
Page 4
“Borg scale” Reference 23 does not mention the Borg scale. As well, reference 30 is from 2000. Please provide a COVID-19 reference to support the use of this scale.
“HADS”—Please provide a COVID-19 related reference to support the use of this 2007 scale.
“aiming for the patient to present a score of between 4 and 5 during training”—please state the full range of the scale so that it is clear to readers where 4 and 5 are situated on the scale.
Page 5
The authors have explained the type of equipment the participants used for their strength training. Was this same equipment available for the control group? Please answer this question in the text.
Please explain in the text the type of low-intensity exercises used during the warm-up.
Please state in the text how the peripheral oximetry oxygen saturation and heart rate were monitored during the program.
Please explain why SPSS v.25 was chosen for the statistical analysis and reference a COVID-19 related publication that used this package in a similar manner.
Beyond the control group having refused participation or they lacked digital literacy, the authors also indicate here that the control group were older than the intervention group and there were significantly more women than men in this group. This must also be stated in the limitations section.
Page 9
“This study validated a supervised 12-week telerehabilitation program carried out with adults with long COVID symptoms at least six months after infection by SARS- COV2.”—please state immediately after this first paragraph why the study was validated. This can be done by moving the second paragraph just before the paragraph beginning “The results of this study can be used to support the use of new technologies”.
Reference 42 is from 2016. The authors are asked to find a COVID-19 related reference to support their claim.
Page 10
Please augment this limitations section as noted above with respect to the differences between the control group and the intervention group.
As per the Instructions for Authors and the information provided as part of the Healthcare Word template, please redo the references to conform to MDPI style.
Page 11
“More Than 50 Long-Term Effects of COVID-19: A Systematic Review and Meta-Analysis.”—this article has now been published. Please reference the published article rather than the preprint.
Page 12
“Kendrick KR, Baxi SC, Smith RM. Usefulness of the modified 0-10 Borg scale in assessing the degree of dyspnea in patients with COPD and asthma. Journal of Emergency Nursing, 26(3), 216-222. https://doi.org/10.1016/S0099-1767(00)90093-X”— year of publication is missing.
Author Response
Thank you very much for your willingness to review the article and for the analysis carried out. as it contributes to the improvement of its content
Só , according to the recommendation:
Page by page suggested edits.
Page 2
Change “A recent report presented by WHO” to “A recent report presented by the World Health Organization (WHO)”.
R: suggested and highlighted change made in the text
“After COVID-19 emerged and caused the collapse of health systems, many patients were not able to receive their treatments because institution-based pulmonary rehabilitation programs were forced to significantly reduce enrollment, or in some cases completely shut down, during the pandemic.”—please provide at least one peer reviewed reference to support the claims of this statement.
added bibliographical reference as suggested
[XXX] Wen J, Milne S, Sin DD. Pulmonary rehabilitation in a postcoronavirus disease 2019 world: feasibility, challenges, and solutions. Curr Opin Pulm Med. 2022;28(2):152-161. doi:10.1097/MCP.0000000000000832
Page 3
Change “the World Health Organization” to “the WHO”.
If the control group included those who refused participation or lacked digital literacy can they truly be said to be a control group? They were, in effect, different from the group who participated in the pilot study in these ways mentioned. The authors need to make this point about the control group in their limitations section.
Yes, we can consider a control group
in the studies
non-equivalent control group, the methodology is clear. we are dealing with a non-randomized control group.
The same health conditions of patients are respected, with the same criteria for intervention. ethics were ensured, giving the opportunity to participate but given the non-existence of conditions, they remained in the control group.
will be added as per the limitations
Page 4
“Borg scale” Reference 23 does not mention the Borg scale. As well, reference 30 is from 2000. Please provide a COVID-19 reference to support the use of this scale.
Thanks for the correction regarding reference 23,
Although reference 30 is from 2000, I believe it to be an extremely relevant article in the use of the Borg scale. new reference added
Gounaridi MI, Vontetsianos A, Oikonomou E, et al. The Role of Rehabilitation in Arterial Function Properties of Convalescent COVID-19 Patients. J Clin Med. 2023;12(6):2233. Published 2023 Mar 13. doi:10.3390/jcm12062233
“HADS”—Please provide a COVID-19 related reference to support the use of this 2007 scale.
Fernández-de-Las-Peñas C, Rodríguez-Jiménez J, Palacios-Ceña M, et al. Psychometric Properties of the Hospital Anxiety and Depression Scale (HADS) in Previously Hospitalized COVID-19 Patients. Int J Environ Res Public Health. 2022;19(15):9273. Published 2022 Jul 29. doi:10.3390/ijerph19159273
“aiming for the patient to present a score of between 4 and 5 during training”—please state the full range of the scale so that it is clear to readers where 4 and 5 are situated on the scale.
added what does 4, 5 mean as suggested
Page 9
Reference 42 is from 2016. The authors are asked to find a COVID-19 related reference to support their claim.
Important article on functional test validation. its placement is explained by this importance
Page 10
Please augment this limitations section as noted above with respect to the differences between the control group and the intervention group.
As per the Instructions for Authors and the information provided as part of the Healthcare Word template, please redo the references to conform to MDPI style.
Page 11
“More Than 50 Long-Term Effects of COVID-19: A Systematic Review and Meta-Analysis.”—this article has now been published. Please reference the published article rather than the preprint.
article published in Natur magazine as identified
Lopez-Leon, S., Wegman-Ostrosky, T., Perelman, C. et al. More than 50 long-term effects of COVID-19: a systematic review and meta-analysis. Sci Rep 11, 16144 (2021). https://doi.org/10.1038/s41598-021-95565-8
Thank you very much for the improvement recommendations. Contributions were introduced from the reference that you indicated in the introduction and discussion section, when talking about the implications for clinical practice.

Round 2
Reviewer 2 Report
Thank you, After revisions i recommend to accept the paper in present form
Author Response
Thank you very much for the review that contributed to improving the investigation
Reviewer 3 Report
Thank you to the authors for the changes they have made to the submission. Those they have made have improved the paper.
There are, however, further changes to be made. A few of these are the result of the new additions that have been made to this version being written by a non-native English speaker as the English is either not able to be understood or too obscure. These statements will need to be rewritten in clear and direct English.
The authors had been asked to redo the references in MDPI style. Of the 49 references, only reference 2 has been redone correctly. Please check the Healthcare Word template for instructions on the required reference style.
Page by page suggested edits.
Page 2
Citation 7 is not to a recent report of the World Health Organization. Furthermore, reference 7 does not provide the information regarding reduced workload and those who did not return to work. Please cite the correct reference.
“caused the collapse of health systems”—A peer reviewed reference is needed to support this claim.
Change “Covid 19” to “COVID-19”.
“did not have great expression and adherence by health professionals”—this statement does not make sense in English. Please have a native English speaker rewrite this in relation to what the authors intended.
Page 4
As references 29 and 30 are both out of date they require supporting references to demonstrate that both the test and the scale are considered relevant for research conducted during COVID-19.
Pages 10-11
Please move the new statement in green that ends with citation 18 to the end of the paragraph, rather than in the middle of it.
Page 11
“It was not possible to assess the degree of customer satisfaction, which could add value to the effective gains with the rehabilitation program.”—this statement in the limitations section is unclear. What do the authors mean by “customer satisfaction”? Who are the customers In this study? If the authors mean the patients, please state this. As well, the authors must explain why it was not possible to determine the patient’s reaction to the intervention.
The authors had been asked by the reviewer in the previous review to add the ways in which the control group differed from the study group to the limitations sections. The authors responded that the differences “will be added as per the limitations”. These additions were not made to the limitations as promised. Please include in the limitations section information on the ways in which the control group was non-equivalent and, thus, a limitation to the study.
The last paragraph of the limitations section does not belong in the Limitations. Please move this to the end of the Discussion.
“The incorporation of telerehabilitation in health systems increases the offer, allowing to respond to all those who do not have access to face-to-face rehabilitation programs and who present conditions of safety and hemodynamic stability feel accompanied. Telerehabilitation is a service of great proximity,”—this is poorly written English. Please have a native English speaker rewrite this information in relation to what the authors intended.
Change “Pandemic” to “pandemic”
Change “Covid-19” to “COVID-19”.
“The beneficial results of the study are the lever in the implementation of personalized telerehabilitation programs, aimed at the person with a complex disease, with a strong focus on the person's assessment and value.”—This statement is obscure. Please have a native English speaker rewrite this information in relation to what the authors intended so that the points made are clear and direct.
Page 13
Reference 37 has no date. Please provide the date.
The concerns regarding the English have been stated in the Comments and Suggestions for Authors.
Author Response
Reply to the Review Report (Reviewer 3)
Dear review:
We appreciate the time you took to review our article and the suggestions made. We try to respond to every one of them.
Page 2
Citation 7 is not to a recent report of the World Health Organization. Furthermore, reference 7 does not provide the information regarding reduced workload and those who did not return to work. Please cite the correct reference.
“caused the collapse of health systems”—A peer reviewed reference is needed to support this claim.
R: Thank you for this observation, giving us the opportunity to correct it.
Change “Covid 19” to “COVID-19”.
R : change made
“did not have great expression and adherence by health professionals”—this statement does not make sense in English. Please have a native English speaker rewrite this in relation to what the authors intended.
We change to: Prior to the COVID-19 pandemic, telemedicine and telehealth didn't gain significant traction or widespread adoption among healthcare professionals. However, the emergence of the pandemic highlighted the need for European authorities to encourage the utilisation of these tools in everyday clinical practice, facilitating their effective implementation without obstacles
Page 4
As references 29 and 30 are both out of date they require supporting references to demonstrate that both the test and the scale are considered relevant for research conducted during COVID-19.
R: Thank you very much for your attention, research articles have been added that provide robustness to the use of the tests applied
Pages 10-11
Please move the new statement in green that ends with citation 18 to the end of the paragraph, rather than in the middle of it.
R: change made
Page 11
“It was not possible to assess the degree of customer satisfaction, which could add value to the effective gains with the rehabilitation program.”—this statement in the limitations section is unclear. What do the authors mean by “customer satisfaction”? Who are the customers In this study? If the authors mean the patients, please state this. As well, the authors must explain why it was not possible to determine the patient’s reaction to the intervention.
The authors had been asked by the reviewer in the previous review to add the ways in which the control group differed from the study group to the limitations sections. The authors responded that the differences “will be added as per the limitations”. These additions were not made to the limitations as promised. Please include in the limitations section information on the ways in which the control group was non-equivalent and, thus, a limitation to the study.
The last paragraph of the limitations section does not belong in the Limitations. Please move this to the end of the Discussion.
R: change made
“The incorporation of telerehabilitation in health systems increases the offer, allowing to respond to all those who do not have access to face-to-face rehabilitation programs and who present conditions of safety and hemodynamic stability feel accompanied. Telerehabilitation is a service of great proximity,”—this is poorly written English. Please have a native English speaker rewrite this information in relation to what the authors intended.
R: change made
Change “Pandemic” to “pandemic”
Change “Covid-19” to “COVID-19”.
R: changes made
“The beneficial results of the study are the lever in the implementation of personalized telerehabilitation programs, aimed at the person with a complex disease, with a strong focus on the person's assessment and value.”—This statement is obscure. Please have a native English speaker rewrite this information in relation to what the authors intended so that the points made are clear and direct.
We change to: The positive outcomes of this study serve as a driving force behind the implementation of personalised telerehabilitation programmes, tailored to individuals dealing with complex diseases. These programmes place a significant emphasis on comprehensive patient assessment and their intrinsic value.
Page 13
Reference 37 has no date. Please provide the date.
R: change made
